# Electrochemical Performance of Photovoltaic Cells Using HDA Capped-SnS Nanocrystal from bis (*N*-1,4-Phenyl-*N*-Morpho-Dithiocarbamato) Sn(II) Complexes

**DOI:** 10.3390/nano10030414

**Published:** 2020-02-27

**Authors:** Johannes Z. Mbese, Edson L. Meyer, Mojeed A. Agoro

**Affiliations:** 1Department of Chemistry, University of Fort Hare, Private Bag X1314, Alice 5700, South Africa; 2Fort Hare Institute of Technology, University of Fort Hare, Private Bag X1314, Alice 5700, South Africa

**Keywords:** single-source precursor, quantum dots, semiconductors, electrochemical, photovoltaic cells

## Abstract

Great consideration is placed on the choice of capping agents’ base on the proposed application, in order to cater to the particular surface, size, geometry, and functional group. Change in any of the above can influence the characteristics properties of the nanomaterials. The adoption of hexadecylamine (HDA) as a capping agent in single source precursor approach offers better quantum dots (QDs) sensitizer materials with good quantum efficiency photoluminescence and desirable particles size. Structural, morphological, and electrochemical instruments were used to evaluate the characterization and efficiency of the sensitizers. The cyclic voltammetry (CV) results display both reduction and oxidation peaks for both materials. XRD for SnS/HDA and SnS photosensitizers displays eleven peaks within the values of 27.02° to 66.05° for SnS/HDA and 26.03° to 66.04° for SnS in correlation to the orthorhombic structure. Current density–voltage (I–V) results for SnS/HDA exhibited a better performance compared to SnS sensitizers. Bode plot results indicate electrons lifetime (τ) for SnS/HDA photosensitizer have superiority to the SnS photosensitizer. The results connote that SnS/HDA exhibited a better performance compared to SnS sensitizers due to the presence of HDA capping agent.

## 1. Introduction

Quantum dots sensitized solar cells (QDSSCs) are emerging innovations for photovoltaic cells as a replacement for the ideal dye-sensitized solar cells (DSSCs). The maximum efficiency of 13.4% obtained from inorganic sensitizer [1] has led many scientists to the research on fabricating better photosensitizes. That can portray a good generation of multiple excitons [2,3], panchromatic characteristics [4], and photostability [5] that will deal with the shortfall of molecular dyes in the traditional DSSCs. The fabrication of quantum dots (QDs) sensitizer materials to enhance this kind of solar cells has gained ground due to their diversity [6,7], tunable band-gap, cost-friendliness, and easy fabrication. QDs size can be controlled to obtain optimum band gap, as well as incorporate other semiconductor QDs materials as co-sensitized to enhance their photospectral properties and increase the conversion efficiency of QDSSCs (as seen in Figure 1) [8,9,10]. Moreover, the challenges involving metal-oxide surface and the absorbers poor contact due to the inabilities of photoinduced charge separation to directly penetrate each other served as a major restriction [11]. 

Other limitation like electrons diffusion length and the electrode nanoporous oxide geometry [12], hinder the surface area optimum adsorption capacity of molecule dye. To increase the cell spectral properties by adopting various photosensitizers will lead to poor optical density [13]. This implies that the amount of solar radiation absorbed by the cells are linked directly to the nanoporous surface area of the electrodes. To solve this shortfall, nanoporous oxide can be separated from absorbers by injection of electrons to improve better overlap with the solar spectrum and higher optical density without destroying the electron absorption performance [14,15,16].

This can be achieved with synthetic process, which is divided into nucleation and growth. The depth knowledge about both steps has resulted in new nanosynthesis route. Giving a better uniform surface morphology, size and monodispersity materials by enabling optimum control on the synthesis process. Factors, such as solvent, reducing agent, and capping agent, are of very great importance in monodisperse nanoparticles synthesis. The use of capping agents in stabilization and colloidal synthesis of nanoparticles is known to control materials size, surface passivation and particle morphology. The adoption of energy-saving and less or non-toxic capping agents will promote green synthetic route of nanoparticle fabrication of large scale commercialization [17,18,19]. Great consideration is placed on the choice of capping agents’ base on the proposed application, in other to cater for the particular surface, size, geometry and functional group. Change in any of the above mention can influence the characteristics properties of the materials. Capping agents, such as trioctylphophine oxide (TOPO), trioctylphophine (TOP), and hexadecylamine (HDA), etc., offer excellent stability with organic solvents for nanoparticles [20,21]. The injection of HDA capping agent could offer desirable particles size and better QDs sensitizer materials with good quantum efficiency. Therefore, enhancing the assembly patterns of the fabricated cells [22]. In the present study, the main objective is exploring the beneficial effects of both HDA capped and uncapped materials on surface treatments. Therefore, leading to improving electrochemical performance of quantum dots sensitizers’ absorber in photovoltaic cells.

## 2. Materials and Methods 

### 2.1. Material 

All materials were purchased and used without modification. The complete test kits containing fluorine-doped tin oxide (FTO) as glass substrate of TiO_2,_ platinum FTO, HI-30 electrolyte iodide, masks, gaskets, chenodeoxycholic acid (CDC) and hot seal were purchased from Solaronix Company (Aubonne, Switzerland). Water, Oleic acid (OA), methanol, HDA, SnS/had, and SnS nanoparticles from bis (*N*-1,4-Phenyl-*N*-Morhpo-dithiocarbamato) Sn(II) complexes.

### 2.2. Synthesis of SnS Nanoparticles with HDA Capping Agent

Nanoparticles were fabricated according to the literature method [23] 0.20 g of bis (*N*-1,4-Phenyl-*N*-Morhpo-dithiocarbamato) tin(II) complex (as seen in Figure 2), was added to 4 mL oleic acid (OA) and were injected into hot HDA of 3 g at 360 °C for surface passivation and particle morphology. 20–30 °C initial temperature was attained for the mixture. The reaction was stabilized at 360 °C and the process lasted for 1 h. The process was allowed to drop to 70 °C signifying the completion of the process and about 50 mL methanol were used the remover of excess OA and HDA. Centrifugation was used to separate the flocculent precipitate and re-dispersed with toluene. Low air pressure was used to remove solvent giving rise to metals sulfides of SnS/HDA nanoparticles. 

### 2.3. Synthesis of SnS Nanoparticles Without HDA Capping Agent

Synthesis of SnS nanocrystals was obtained through high-temperature thermal decomposition of bis (*N*-1,4-Phenyl-*N*-Morhpo-dithiocarbamato) tin(II) complex using Perkin Elmer TGA 4000 ThermoGravimetric Analyser (TGA) (San Jose, CA, USA). About 25 mg of the complex was loaded into an alumina pan and weight changes were recorded as a function of temperature for a 10 °C min^−1^ temperature gradient between 30–900 °C. A purge gas of flowing nitrogen at a rate of 20 mL min^−1^ was used. At temperatures between 360 and 900 °C, the complex end product was converted into residue, which was expected for the formation of SnS nanocrystals from the residue obtained from the TGA. 

### 2.4. Fabrication and Assembling of Solar Cells

QDSSC were prepared with 2 × 2 cm^2^ FTO-glass plates of platinum and TiO_2_ electrodes were purchase from Solaronix (Aubonne, Switzerland) with 6 × 6 mm^p^ active areas of TiO_2_ screen coated. Dye loading for sensitization was done using 10 mL of warm water with MO-SnS/HDA and MO-SnS. Co-adsorbents were added (Co-adsorbent/dye) of chenodeoxycholic acid (CDC) were used. The mediating solution was the commercial HI-30 electrolyte solution (Solaronix), with content of iodide species at 0.05 M. The TiO_2_ thin film was soaked into a solution of photosensitizers for 24 h. The two substrates, one coated with TiO_2_ loaded with photosensitizers and the other with platinum, were held together using polyethylene and soldering iron. The syringe was used to inject the HI-30 electrolyte (iodide). 

Characterization and solar cell conversion efficiency (%) measurements were done using Equations (1)–(4) mentioned below. 

All substitutions would be done following Equations (1)–(4).
Jsc = (Isc (mA))/(A(cm^2^))(1)
FF = (V_MAX_ × J_MAX_)/(V_OC_ × J_SC_)(2)
Where FF is fill factor, and V_OC_ is an open circuit voltage; J_SC_ is a short circuit current density.
η (%) = (V_MAX_ × J_MAX_)/P*in*.(3)

The above equation was adopted where P*in* is a light intensity of 100 mW cm^−2^.
η (%) = (V_OC_ × J_SC_ × FF)/P*in* × 100%.(4)

The final equation for overall conversion efficiency was derived from the following equation.

### 2.5. Physical Measurements

Electrochemical studies were carried out using Metrohm 85695 Autolab with Nova 1.10 software (Metrohm South Africa (Pty) Ltd., Sandton, South Africa). A platinum electrode was adopted as a counter electrode, with TiO_2_ as the anode, while HI-30 iodode electrode was used as a reference electrode. Cyclic voltammetry (CV) was performed at scan rates from 0.05 to 0.35 V s^−1^ with an increment of 0.05 V s^−1^. All the experiments were performed at room temperature. Electrochemical impedance spectroscopy (EIS) was carried out in the frequency range of 100 kHz to 100 mHz. Current density–voltage (I–V) parameters were collected through a Keithley 2401 source meter and a Thorax light power meter. Lumixo AM1.5 light simulator (RS Components (SA), Midrand, South Africa) was employed, and the lamp was fixed at 50 cm high to avoid illumination outside of the working area. To avoid cells degradation, temperature was kept below 60 °C and the light power density at 100 mW cm^−2^ (AM1.0). X-ray diffractometer (XRD) were employed to evaluate the structural pattern of the samples, and the diffraction structure was obtained between 10 and 90° at the interval of 0.05°. The surface roughness of the SnS/HDA and SnS FTO substrates were identified through the use of atomic force microscopy (AFM) (JPK NanoWizard II AFM, JPK Instruments, Berlin, Germany) at a scan rate of 0.8 Hz in contact mode. JEOL JEM 2100 High-Resolution Transmission Electron Microscope (JEOL Inc., Pleasanton, CA, USA) (HRTEM) operating at 200 KV with selected area electron diffraction (SAED) patterns was used.

## 3. Results and Discussion

SnS/HDA and SnS sensitizers were employed to investigate the optimized electrochemical performance of both materials. Figure 3 shows comparative CV curves of two the materials at fix applied amplitude of 50 mV s^−1^. The measurement shows precise redox peaks at the voltage of 0.0 and 0.6 V, confirming the charge storage by redox reactions. The kinetic irreversibility of both displaced materials has asymmetry, i.e., reduction and oxidation of the CV curves [24,25]. SnS sensitizer exhibited a higher area than SnS/HDA, which is preferred for higher performance abilities.

Based on the EIS results of SnS/HDA and SnS photosensitizer, shown in Figure 4, the charge transfer resistance (*R_ct_*) of the SnS sensitizer film decreases compare to the SnS/HDA sensitizer with lower charge transfer resistivity. These will enhance fast electron transfer, lower charge recombination, and better conductivity compared to SnS/HDA photosensitizer. This connotes improved contact between the redox couple with SnS sensitizer and a better degree of electron growth [26,27]. By this, we concluded that SnS sensitizer offers preferable efficient electron transfer.

The appropriate instrument to investigate structural phase unit cell dimension and lattice parameters of SnS/HDA and SnS photosensitizer films is X-ray diffraction (XRD) analysis. Figure 5 reveals the XRD configurations. Their peaks of 2*θ* range between 27–82° that correlated to orthorhombic SnS crystals (JCPDS 039-0354). Furthermore, both materials display no traces of other impurities as observed in the XRD patterns. The purity and quality of the crystalline are displayed by the strong and sharp diffraction peaks observed in both samples. The improvement in crystalline quality could be linked to nucleation control, which promotes the growth process of SnS [28]. In addition, there is no change in the preferential orientation along (201), (210), (111), (301), (311), (511), (610), and (512) for both samples, which confirms successful fabrication of SnS and SnS/HDA into the lattice of TiO_2_ [29]. The preferential orientation of SnS/HDA is linked to the nucleation control of the growth process due to excess HDA used during the synthesis [30]. The presence of lower intensities and minor phases are usually linked to defect chemistry, thermodynamics of solid, and thermal gradient.

Table 1 and Figure 6 display I–V parameters of SnS/HDA and SnS photosensitizers quantum dot solar cells (QDSCs). TiO_2_ photoanode sensitized with the fabricated dye, and Pt counter electrode and redox couple HI-30 electrolyte was used. In order to evaluate the efficiency of the photosensitized materials, these materials were fabricated with the same test conditions. The SnS/HDA and SnS JSC is observed at 10.99 mA cm^−2^ and 1.802 mA cm^−2^, producing conversion efficiency at 1.25% and 0.42%. Their photovoltaic characteristics of QDSSCs films based for V_OC_ are 0.423 V and 0.375 V. The lower conversion efficiency of SnS compared to SnS/HDA could be linked to the low electrocatalytic activity, which reduce HI-30 iodide and low electrical conductivity. These hinder electron transport from recombining with holes because of redox couple [31].

The Bode plot, as seen in Figure 7, exhibits the phase angle shift and frequency response magnitude. According to Bode plot, the phase angle of SnS photosensitizer is higher compared to SnS/HDA photosensitizer. This depicts that SnS has a porous nature (confirmed by surface area analysis and AFM image) and lower charge transfer resistance. Furthermore, this enhances the injection and flow of electrolyte in the cell [32]. 

The topographical analysis was carried out using atomic force microscope (AFM). The evaluated particles size varied in the range of 0.11–1.18 um for SnS/HDA and 0.054–0.54 um for SnS films grown at 360 °C with size height of 16.8% and 8.4%, respectively, as seen in Figure 8. These images revealed little significant changes in the particles size of both samples fabricated at the same temperature. The ad-atoms move over the surface brings bonding togetherness with an enlarged form of sub-particles. This emanated to formation good compactness and larger particles size. When the film surface is smooth, it resulted to lower roughness due to high surface diffusion [33]. This can be attributed to better particle size and good power conversion efficiency in QDSSCs, which depends solely on the low roughness of the photoelectrodes. 

The measurements and morphology of the synthesized SnS/HDA and SnS were examined by HRTEM; Figure 9 reveals, that the products are spherical, clustered, and agglomerated, with crystallite sizes within the range 14.96–44.39 nm for SnS and 9.5–23.19 nm for SnS/HDA [34]. Their lattice fringes revealed polycrystalline nanoparticles comprising numerous crystal grains d-spacing of 3.549–3.623 nm, affirming their polycrystalline nature, which is in concurrence with the report from Reference [35]. SAED obtained reveals the spot pattern shaping of the diffraction rings, which indicates that both materials have high crystalline nature [36].

## 4. Conclusions

In the CV curve of SnS/HDA and SnS sensitizers, the spectra displaced both reduction and oxidation peaks for both materials. The diffusion of SnS/HDA and SnS photosensitizer in HI-30 electrolyte connote Warburg’s constant (W) as a result of the straight line. The SnS/HDA sensitizer displaced lower impedance compare to SnS sensitizer. XRD results for SnS/HDA and SnS photosensitizers displaced eleven peaks within the values of 27.02° to 66.05° for SnS/HDA and 26.03° to 66.04° for SnS in correlation to orthorhombic structure. The I–V efficiency obtained indicates that the SnS/HDA exhibited a better performance compared to SnS and Sn(II) sensitizers due to the presence of HDA capping agent. Bode plot results indicate that the electrons lifetime (τ) for SnS/HDA photosensitizer displaced superiority to the SnS photosensitizer, owning to their ability to enhanced electron lifetime and reduced electron recombination. The AFM results show the particle size distribution in SnS value at 357 nm with a smooth surface and good compactness on the substrate. However, the value for SnS/HDA is 122 nm displaced shape and size of non-symmetrical particles. The lattice fringes revealed polycrystalline nanoparticles comprising numerous crystal grains for both samples.

## Figures and Tables

**Figure 1 nanomaterials-10-00414-f001:**
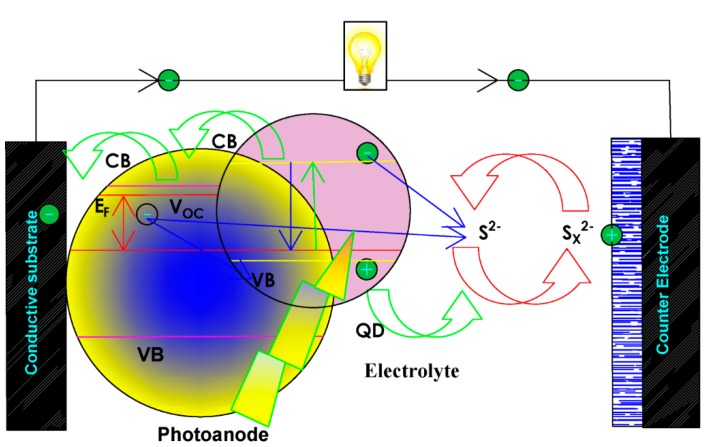
Schematic illustration of quantum dot solar cells (QDSCs) working procedure involving, quantum dots (QDs) photosensitizer, counter electrode, photoanode, and electrolyte [12].

**Figure 2 nanomaterials-10-00414-f002:**
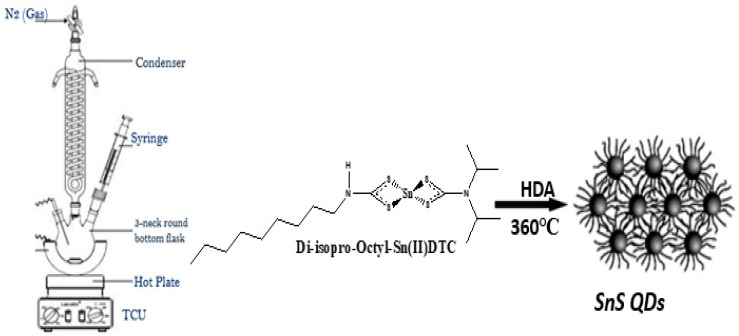
Schematic reaction of nanoparticles.

**Figure 3 nanomaterials-10-00414-f003:**
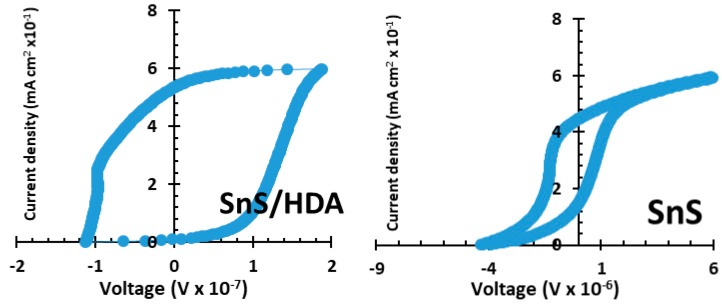
Cyclic voltammetry (CV) spectra of SnS/HDA and SnS nanoparticles.

**Figure 4 nanomaterials-10-00414-f004:**
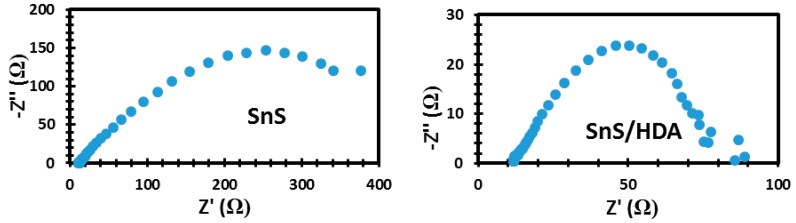
Electrochemical impedance spectroscopy (EIS) spectra of SnS/HDA and SnS nanoparticles.

**Figure 5 nanomaterials-10-00414-f005:**
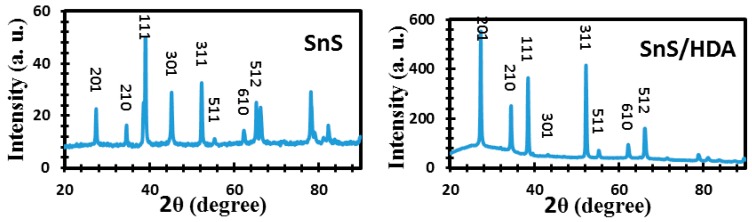
XRD spectra of SnS/HDA and SnS nanoparticles.

**Figure 6 nanomaterials-10-00414-f006:**
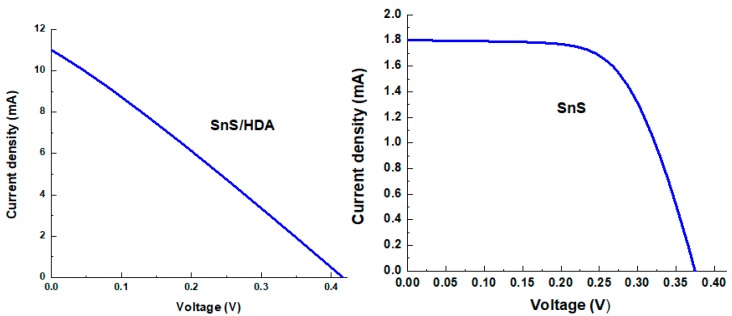
I–V curve characteristics of SnS/HDA and SnS nanoparticles.

**Figure 7 nanomaterials-10-00414-f007:**
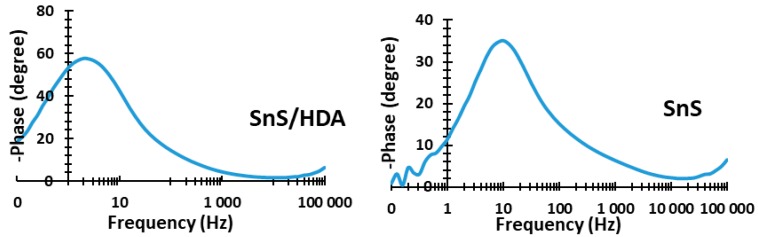
Bode plot spectra of SnS/HDA and SnS nanoparticles.

**Figure 8 nanomaterials-10-00414-f008:**
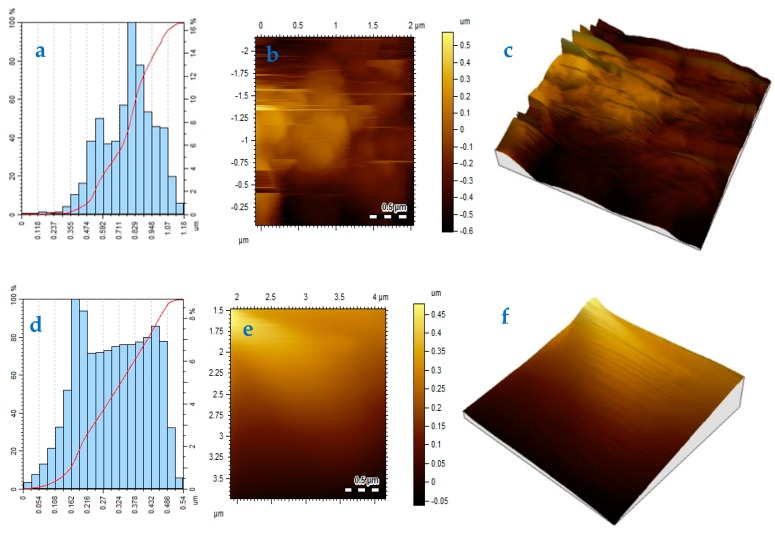
Height profile (**a**,**d**), 2D (**b**,**e**), and 3D (**c**) atomic force microscopy (AFM) images of SnS (**a**–**c**) and SnS/HDA (**d**–**f**) nanoparticles.

**Figure 9 nanomaterials-10-00414-f009:**
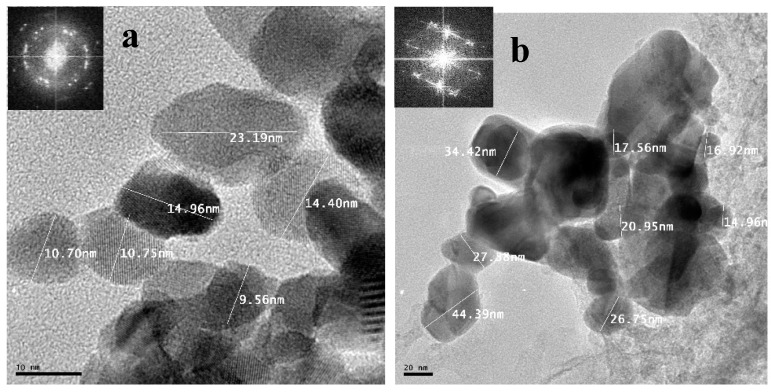
High-Resolution Transmission Electron Microscope (HRTEM) images of (**a**) SnS/HDA and (**b**) SnS nanoparticles.

**Table 1 nanomaterials-10-00414-t001:** I–V characteristics of SnS/HDA and SnS nanoparticles.

Dye	Photoanode	Electrolyte	CEs	J_SC_ (mA/cm^2^)	V_OC_ (mV)	FF	η (%)
**SnS/HDA**	TiO_2_	HI-30	Pt	10.99	0.423	0.27	**1.25**
**SnS**	TiO_2_	HI-30	Pt	1.802	0.375	0.63	**0.42**

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
