# Peer review of "Electrochemical Performance of Photovoltaic Cells Using HDA Capped-SnS Nanocrystal from bis (*N*-1,4-Phenyl-*N*-Morpho-Dithiocarbamato) Sn(II) Complexes"

_nanomaterials, 2020, doi:10.3390/nano10030414_

Round 1
Reviewer 1 Report
This work studied the effect of capping ligand on the working performance of SnS-QDSSC. All the results in the manuscript well manifest the SnS-QDSSC with capping HDA shows better efficiency, and the authors suggest the morphological effect by capping HDA plays an important role. In this regard, this study is suitable for publication, but the following points should be clarified.
(1) In Figure 5, the main peaks are observed in the sample degree, but there are more peaks in SnS/HDA. Also, their intensities are quite a difference. More explanation is required for these difference.
(2) For the authors' suggestion of morphological effect, additional measurements such as TEM will give much clearer evidence.
(3) As a minor point, some abbreviations appear, but their full names are explained in the middle of paper.
Reviewer 2 Report
The manuscript entitled "Electrochemical Performance of Photovoltaic Cells using HDA Capped-SnS Nanocrystal from bis(N-1,4-Phenyl-N-Morpho-dithiocarbamato Sn(II) complexes" has described a method based on SNS/HDA to increase the efficiency of a solar cell. After reviewing this paper, here are suggestions that I recommend authors can consider to improve:
Why can SNS/HDA improve the solar cell efficiency? The authors did not clearly demonstrate the main mechanism. The quality of equations needs to improve. (page 4) Theoretical explanation is needed.
Attached, please see the paper with my comments for your reference.
